# Acculturation and Subjective Norms Impact Non-Prescription Antibiotic Use among Hispanic Patients in the United States

**DOI:** 10.3390/antibiotics12091419

**Published:** 2023-09-08

**Authors:** Lindsey A. Laytner, Kiara Olmeda, Juanita Salinas, Osvaldo Alquicira, Susan Nash, Roger Zoorob, Michael K. Paasche-Orlow, Barbara W. Trautner, Larissa Grigoryan

**Affiliations:** 1Department of Family and Community Medicine, Baylor College of Medicine, Houston, TX 77098, USA; 2Center for Innovations in Quality, Effectiveness and Safety (IQuESt), Houston, TX 77021, USA; 3Tilman J. Fertitta Family College of Medicine, Houston, TX 77021, USA; 4Department of Medicine, Tufts Medical Center, Boston, MA 02111, USA; 5Michael E. DeBakey Veterans Affairs Medical Center, Houston, TX 77030, USA; 6Department of Medicine, Section of Health Services Research, Baylor College of Medicine, Houston, TX 77030, USA

**Keywords:** acculturation, subjective norms, socio-cultural factors, antibiotic resistance, non-prescription antibiotic use, antibiotic stewardship

## Abstract

Using antibiotics without medical guidance (non-prescription antibiotic use) may contribute to antimicrobial resistance. Hispanic individuals are a growing demographic group in the United States (US) with a high prevalence of non-prescription antibiotic use. We investigated the effects of acculturation and subjective norms on Hispanic individuals’ intentions to use antibiotics without a prescription from the following sources: (1) markets in the United States (not legal), (2) other countries (abroad), (3) leftovers from previous prescriptions, and (4) friends/relatives. We surveyed self-identified Hispanic outpatients in eight clinics from January 2020 to June 2021 using the previously validated Short Acculturation Scale for Hispanics (SASH). Of the 263 patients surveyed, 47% reported previous non-prescription use, and 54% expressed intention to use non-prescription antibiotics if feeling sick. Individuals with lower acculturation (Spanish-speaking preferences) expressed greater intentions to use antibiotics from abroad and from any source. Individuals with more friends/relatives who obtain antibiotics abroad were over 2.5 times more likely to intend to use non-prescription antibiotics from friends/relatives (*p* = 0.034). Other predictors of intention to use non-prescription antibiotics included high costs of doctor visits and perceived language barriers in the clinic. Antibiotic stewardship interventions in Hispanic communities in the United States should consider the sociocultural and healthcare barriers influencing non-prescription use and promote language-concordant healthcare.

## 1. Introduction

Using antibiotics without a prescription (non-prescription antibiotic use) is a common practice worldwide and is a safety threat to individuals and the public health [1,2,3]. Non-prescription antibiotic use can potentially increase the risks of adverse drug reactions or interactions, superinfection, gut dysbiosis, and the development of antimicrobial resistance [4,5,6].

Recent studies have documented the determinants of non-prescription antibiotic use across low-, middle-, and high-income countries and found that patient-level (sociocultural and sociodemographic) factors and healthcare system barriers contribute to non-prescription antibiotic use [7,8,9]. In the United States (US), Hispanic communities have one of the highest reported prevalence rates of non-prescription antibiotic use, with the prevalence ranging from 19 to 66% [9]. Prior studies have identified that these Hispanic communities use non-prescribed antibiotics from a variety of sources, including leftover prescriptions (e.g., from self, friends, or family); purchasing illegally under-the-counter through informal sources in the US (e.g., flea markets and ethnic or herbalist shops); or outside the US without a prescription (e.g., across the border in another country, including Mexico) [3,9,10,11,12].

Sociocultural factors include an individual’s level of acculturation and their subjective norms. Acculturation and subjective norms can impact health behaviors in Hispanic communities [10,11,12]. Acculturation is “the process by which individuals adopt the attitudes, values, customs, beliefs, and behaviors of another culture” [13,14]. For example, less-acculturated individuals may continue certain health practices (e.g., non-prescription antibiotic use) that they had in their home countries. Subjective norms are often classified as the “expectations set by groups of important people (such as family, relatives, and friends) in terms of whether an individual should or should not engage in a behavior” [15]. For instance, individuals may engage in non-prescription antibiotic use because their friends and family also routinely engage in that behavior.

A recent study of a Hispanic community along the Texas border found that a higher generation score, a proxy measure of acculturation, was associated with lower cross-border purchases of antibiotics [11]. Another qualitative study of Hispanic primary care patients in Houston found that patients’ subjective norms (e.g., friends and family frequently purchase non-prescription antibiotics) and social networks (e.g., friends, family, or other “trusted” persons) influenced their decisions to use non-prescription antibiotics [12]. However, these studies included medically underserved, impoverished Hispanic individuals and did not study the independent effects of sociocultural and healthcare system factors on non-prescription antibiotic use in sociodemographically diverse Hispanic communities.

## 2. Materials and Methods

We investigated the effects of Hispanic patients’ sociocultural factors (acculturation and subjective norms) and the barriers to healthcare on the intention to use non-prescription antibiotics from four sources: (1) markets in the United States (under the counter, not legal), (2) other countries, (3) leftovers from previous prescriptions, and (4) friends/relatives.

### 2.1. Design and Recruitment

We conducted a large, cross-sectional survey to assess non-prescription antibiotic use in sociodemographically diverse outpatients. Data collection occurred between January 2020 and June 2021 in eight outpatient clinic waiting rooms (six public primary care and two private emergency departments) in Harris County, Texas [3]. Clinic staff gave flyers to patients who checked in for primary care visits. The flyer summarized the study, and interested patients volunteered to participate. Surveys were conducted anonymously in person when permitted during the pandemic or remotely via teleconferencing in patients’ preferred language (English or Spanish). Each respondent was given a list of brand name and generic antibiotics that were accompanied by images of the most commonly used antibiotics in the US and Latin American countries.

Individuals who self-identified as Hispanic or Latino in the larger survey were included for analysis in this study [3]. This study was approved by the Institutional Review Board for Baylor College of Medicine and Affiliated Hospitals (protocol H-45709). Additional details on recruitment, survey design information, sample size calculations, response rate, and additional information were published elsewhere [3].

### 2.2. Survey Instrument

The survey instrument is available in Appendix A. Non-prescription antibiotic use was defined as the consumption of antibiotics not prescribed to that individual for his or her current health condition [3]. Intended use was defined as a professed intention for future non-prescription antibiotic use [9,16]. Individuals classified as non-prescription antibiotic users reported having “ever taken” oral antibiotics without a prescription. Individuals classified as intended users endorsed using antibiotics from one of four sources presented via the question “If you were feeling sick, would you take antibiotics in the following situations without contacting a doctor/nurse/dentist/clinic?” The sources presented to the patients were: (1) you can buy antibiotics without a prescription in the United States, (2) you can buy antibiotics without a prescription in another country, (3) friends or relatives give you antibiotics, and (4) you have leftover antibiotics from a previous prescription.

Survey questions were mapped to factors in the Kilbourne Framework for Advancing Health Disparities Research, including the patient, healthcare system, and clinical encounter factors [17]. Subjective norms and acculturation are patient factors that may contribute to non-prescription antibiotic use in Hispanic populations. Healthcare system barriers included lacking transportation to the doctor visits, long clinic waits, not having a regular doctor, and the high cost of doctor visits. Clinical encounter factors included language barriers at the clinic and during doctor visits.

### 2.3. Patient Factors

#### 2.3.1. Sociodemographic Factors

Sociodemographic characteristics included age, gender, race/ethnicity, education, yearly household income, health insurance status, country of birth, and health literacy (Table 1). Individuals’ insurance status was categorized into three groups: (1) private insurance or Medicare, (2) Medicaid (i.e., public health insurance for low-income children, families, seniors, and people with disabilities) or county financial assistance program (CFAP) (provides healthcare coverage/access to publicly funded clinics at very low or no cost to patients), or (3) self-pay (no insurance or CFAP). For health literacy, we used the Brief Health Literacy Screen measure validated in primary care settings [18,19]. Inadequate health literacy was defined as an answer to any of three screening questions that endorsed having problems associated with health literacy some or all of the time.

#### 2.3.2. Acculturation

Acculturation was assessed using the Short Acculturation Scale for Hispanics (SASH) developed and validated by Marin et al. and included the Language Use, Television and Media, and Social and Ethnic Relations subscale scores [20]. The SASH questionnaire contains 12 questions of equal weight: 5 assessing language preferences, 3 assessing media preferences, and 4 assessing social and ethnic preferences (Appendix A. Survey Instrument).

Each question was scored on Likert-type scales, ranging from 1 (Only Spanish/all Hispanic individuals) to 5 (Only English/all non-Hispanic individuals). Subscores with lower values (closer to 1) reflect preferences for Spanish-speaking interactions or Hispanic social interactions or entertainment. Higher scores (closer to 5) reflect a preference for English-speaking interactions or entertainment. The total points per subscale were averaged over the number of questions answered to generate the numerical score for that subscale. Cronbach’s alpha was computed to determine the internal consistency of the set of questions representing each construct; one question was excluded from the ethnic and social relations subscale (Appendix B. SASH Reliability Statistics). The overall acculturation score was calculated using the sum of the language, television and media use, and ethnic and social relations subscores (Table 2).

#### 2.3.3. Subjective Norms

Subjective norms were identified by the following proxy questions: (1) “How many of your friends or relatives use antibiotics without contacting a doctor?” and (2) “How many of your friends or relatives get antibiotics from another country?” Each question was scored using Likert-type scales ranging from 1 (none/don’t remember/don’t know) to 4 (all/most/about half) (Table 3).

### 2.4. Healthcare System and Clinical Encounter Factors

Healthcare barriers were assessed using five questions relevant to our safety net patient population, including transportation, language barriers, long clinic waits, not having a regular doctor, and the high cost of doctor visits. Each question was scored dichotomously as “not a problem” or “a problem” (i.e., included answering that the barrier was a minor or major problem) (Appendix C).

### 2.5. Statistical Analysis

Descriptive statistics were performed on all study variables using SPSS version 28 (Chicago, IL, USA) [21]. Cronbach’s alpha was computed to analyze the internal consistency of the set of questions representing the acculturation scale. We used univariate logistic regression to assess the patient and healthcare system factors associated with patients’ intention to use non-prescription antibiotics from each source. Predictor variables that showed a univariate relationship (*p* < 0.2) with each source of intended non-prescription antibiotic use were considered for the multivariate analyses (Appendix D. Univariate Regression Results).

Multivariate logistic regression assessed the effects of patients’ acculturation and subjective norms on their intention to use antibiotics without a prescription from one of the following sources: (1) stores or markets in the US, (2) another country, (3) friends/relatives, (4) a leftover prescription, and (5) any of these four sources (Table 4, Table 5, Table 6, Table 7 and Table 8).

## 3. Results

### 3.1. Patient Factors

#### 3.1.1. Sociodemographic Characteristics

Table 1 shows the sociodemographic characteristics of the 263 patients surveyed. Most respondents were female (74%) and educated at the high school level (40%) or some college and above (29%). Approximately 61% of the patients had healthcare coverage through Medicaid or county financial assistance, followed by private insurance or Medicare (28%) and self-pay (6%). Most patients who reported their income had household incomes below 40,000 USD/year (70%). Approximately 59% of all patients preferred being surveyed in Spanish. More patients were born outside the US (69%) in Mexico (n = 131), followed by Honduras (n = 15) and El Salvador (n = 14). Foreign-born patients lived in the US for a median of 23.5 years. Nearly half (47%) of the participants reported prior non-prescription antibiotic use, and over 75% of patients were classified as having “adequate” health literacy levels (Table 1). Overall, 54% professed an intention to use/obtain non-prescription antibiotics if feeling sick from at least one source (Figure 1).

#### 3.1.2. Acculturation

Table 2 includes the mean acculturation subscale and Cronbach’s alpha scores for language use, TV and media, ethnic and social relations, and the total acculturation (overall, aggregate score). The median total acculturation score for respondents who professed intention for future non-prescription antibiotic use was 2.5/5.0 (IQR 1.6–3.4), and the Cronbach’s alpha for the total acculturation was 0.939 (Table 2).

#### 3.1.3. Subjective Norms

Table 3 lists the proportion of patients (N = 263) that reported friends/relatives that used non-prescribed antibiotics or purchased antibiotics from other countries (outside the US). Over 60% of patients reported that some to all of their friends or relatives had used antibiotics without contacting a doctor. About 53% reported that some to most of their friends or relatives have used or purchased non-prescribed antibiotics from another country (Table 3).

### 3.2. Healthcare System Factors

Appendix C displays the patient-reported barriers to healthcare in the last 12 months. Of the barriers to access, patients frequently expressed that long waiting times (26%), transportation (16%), and the high cost of doctor visits (16%) were problematic, followed by language barriers (8%) and not having a regular doctor (5%) (Appendix C).

### 3.3. Multivariate Results

Table 4, Table 5, Table 6, Table 7 and Table 8 display the multivariate logistic regression results of patient intentions to use non-prescription antibiotics from each of the four sources and overall.

**Table 4 antibiotics-12-01419-t004:** Multivariate results of the intended use of antibiotics from stores or markets in the US.

Intended Use of Antibiotics from Stores or Markets in the United States		
Predictors ^§^	OR (95% CI)	*p-*Value
Prior Non-prescription Use		
No Prior Use	1 (reference)	1 (reference)
Prior Use	6.26 (3.13–12.51)	<0.001
Barriers To Healthcare Access For your medical appointments in the last 12 months, how much of a problem are: High cost of doctor visits		
Not a problem	1 (reference)	1 (reference)
A problem	3.1 (1.43–6.69)	0.004

^§^ The following predictors were not significant in the multivariate model: Acculturation (Language Use Subscale, Media Subscale, and Ethnic Social Relations Subscale); Subjective Norms (How many of your friends or relatives get antibiotics without contacting a doctor? How many of your friends or relatives get antibiotics from another country?); Sociodemographics (Age, Healthcare System, Insurance, Language, Education, and Country of Birth); and Barriers to Healthcare Access (For your medical appointments in the last 12 months, how much of a problem is not having a regular doctor?).

**Table 5 antibiotics-12-01419-t005:** Multivariate results of the intended use of antibiotics bought without a prescription from another country.

Intended Use of Antibiotics Bought without a Prescription from Another Country		
Predictors ^§^	OR (95% CI)	*p*-Value
Acculturation ^¶^		
Ethnic Social Relations Subscale	0.54 (0.33–0.86)	0.009
Prior Non-prescription Use		
No Prior Use	1 (reference)	1 (reference)
Prior Use	10.49 (5.13–21.46)	<0.001

^¶^ The Short Acculturation Scale for Hispanics is comprised of three subscales (Language Use, Media, and Ethnic Social Relations). Scores range from 1 (all Latinos/Hispanics) to 5 (all non-Latinos/Hispanics). Higher score indicates higher levels of acculturation [20]. ^§^ The following predictors were not significant in the multivariate model: Acculturation (Language Use Subscale and Media Subscale); Subjective Norms (How many of your friends or relatives get antibiotics without contacting a doctor? How many of your friends or relatives get antibiotics from another country?); Sociodemographics (Age, Years lived in the US, and Education); and Barriers to Healthcare Access (For your medical appointments in the last 12 months, how much of a problem are high cost of doctor visits, a language barrier, and not having a regular doctor?).

**Table 6 antibiotics-12-01419-t006:** Multivariate results of the intended use of antibiotics from friends and relatives.

Intended Use of Antibiotics from Friends and Relatives		
Predictors ^§^	OR (95% CI)	*p*-Value
Social Norms		0.054
How many of your friends or relatives get antibiotics from another country?		
None	1 (reference)	1 (reference)
Some	2.52 (0.92–6.93)	0.072
All/Most/About Half	2.51 (1.07–5.85)	0.034
Don’t Know/Don’t remember	0.95 (0.34–2.64)	0.918
Sociodemographics		
Education		0.017
Less than High School	1 (reference)	1 (reference)
High school or GED	0.53 (0.24–1.19)	0.126
Some College or Above	0.32 (0.14–0.7)	0.004
Prior Non-prescription Use		
No Prior Use	1 (reference)	1 (reference)
Prior Use	10.59 (5.0–22.43)	<0.001
Barriers To Healthcare Access		
For your medical appointments in the last 12 months, how much of a problem are:		
High cost of doctor visits		
Not a problem	1 (reference)	1 (reference)
A problem	3.16 (1.38–7.21)	0.006

^§^ The following predictors were not significant in the multivariate model: Acculturation (Language Use Subscale, Media Subscale, and Ethnic Social Relations Subscale); Subjective Norms (How many of your friends or relatives get antibiotics without contacting a doctor?); Sociodemographics (Age, Sex, Insurance, and Country of Birth); and Barriers to Healthcare Access (For your medical appointments in the last 12 months, how much of a problem are transportation, a language barrier, and not having a regular doctor?).

**Table 7 antibiotics-12-01419-t007:** Multivariate results of the intended use of antibiotics from leftover antibiotic courses.

Intended Use of Antibiotics from Leftover Courses		
Predictors ^§^	OR (95% CI)	*p*-Value
Prior Non-prescription Use		
No Prior Use	1 (reference)	1 (reference)
Prior Use	7.51 (4.15–13.26)	<0.001
Barriers To Healthcare Access		
For your medical appointments in the last 12 months, how much of a problem are: A language barrier		
Not a problem	1 (reference)	1 (reference)
A problem	3.08 (1.03–9.26)	0.006

^§^ The following predictors were not significant in the multivariate model: Acculturation (Language Use Subscale, Media Subscale, and Ethnic Social Relations Subscale); Subjective Norms (How many of your friends or relatives get antibiotics without contacting a doctor? How many of your friends or relatives get antibiotics from another country?); Sociodemographics (Healthcare System, Insurance, and Country of Birth); and Barriers to Healthcare Access (For your medical appointments in the last 12 months, how much of a problem are long waiting times or the high cost of doctor visits?).

**Table 8 antibiotics-12-01419-t008:** Multivariate results of the intended use of antibiotics from any source (US, abroad, friends and relatives, and leftover courses).

Intended Use of Antibiotics from Any Source (US, Abroad, Friends and Relatives, and Leftover Courses)		
Predictors ^§^	OR (95% CI)	*p*-Value
Acculturation ^¶^		
Language Use Subscale	0.61 (0.39–0.96)	0.031
Sociodemographics		
Country of Birth		
Born in US	1 (reference)	1 (reference)
Born in other countries ^‖^	8.47 (2.56–28.02)	<0.001
Prior Non-prescription Use		
No Prior Use	1 (reference)	1 (reference)
Prior Use	12.32 (6.58–23.09)	<0.001

^¶^ The Short Acculturation Scale for Hispanics is comprised of three subscales (Language Use, Media, and Ethnic Social Relations). Scores range from 1 (all Latinos/Hispanics) to 5 (all non-Latinos/Hispanics). Higher score indicates higher levels of acculturation [20]. ^‖^ Includes 1 Columbia, 1 Costa Rica, 6 Cuba, 1 Dominican Republic, 14 El Salvador, 6 Guatemala, 15 Honduras, 131 Mexico, 2 Nicaragua, 1 Panama, 1 Peru, and 3 Venezuela (countries are listed in alphabetical order). ^§^ The following predictors were not significant and therefore not included in this model: Acculturation (Media Subscale and Ethnic Social Relations Subscale); Social Norms (How many of your friends or relatives get antibiotics from another country?); Sociodemographics (Education); and Barriers to Healthcare Access (For your medical appointments in the last 12 months, how much of a problem are the high cost of doctor visits and a language barrier).

#### 3.3.1. Intended Use of Antibiotics from Stores or Markets in the United States

The high costs of doctor visits (OR 3.1, 95% CI [1.43–6.69], *p* = 0.004) and prior non-prescription antibiotic use (OR 6.3, 95% CI [3.13–12.51], *p* < 0.001) were significant predictors of the intended use of non-prescribed antibiotics purchased in the US. Neither the acculturation subscales nor subjective norms were significant predictors of the intended use of non-prescription antibiotics from stores or markets in the US (Table 4).

#### 3.3.2. Intended Use of Antibiotics Bought without a Prescription from Another Country

Individuals with lower Ethnic and Social Relations subscale scores indicating higher preferences to socialize and associate with other Hispanic individuals had higher odds of the intention to use non-prescribed antibiotics from another country compared to those with higher Ethnic and Social Relations subscale scores (OR 0.54 95% CI [0.33–0.86], *p* = 0.009). In addition, patients with prior non-prescription antibiotic use had 10.5 times higher intended use from other countries (95% CI [5.13–21.46], *p* < 0.001) (Table 5).

#### 3.3.3. Intended Use of Antibiotics from Friends and Relatives

More educated patients with a high school, college, or above education were 68% less likely to use antibiotics from a friend or relative (OR 0.32, 95% CI [0.14–0.70], *p* = 0.004). The high cost of doctor visits during medical appointments was a significant predictor of intention (OR 3.16, 95% CI [1.38–7.21], *p* = 0.006). Patients who reported at least some of their friends or relatives getting non-prescribed antibiotics from other countries had 2.5 times higher odds of intention to use non-prescription antibiotics from friends and relatives (95% CI [1.07–5.85], *p* = 0.034). Additionally, patients with prior non-prescription antibiotic use had over 10.6 times higher odds of intended use from friends and relatives (95% CI [5.0–22.4], *p* < 0.001) (Table 6).

#### 3.3.4. Intended Use of Antibiotics from Leftover Courses

Patients reporting language barriers as a problem during their medical appointment had over three times higher odds of the intention to use antibiotics from leftover prescription sources than patients who did not (95% CI [1.03–9.26], *p* = 0.006). Prior non-prescription antibiotic use was a strong predictor of the intention from leftover courses (OR 7.5 95% CI [4.15–13.26], *p* < 0.001). Neither acculturation subscales nor social norms were significant predictors of the intended use from leftover antibiotic courses (Table 7).

#### 3.3.5. Intended Use of Antibiotics from Any Source (US, Abroad, Friends and Relatives, and Leftover Courses)

Patients born outside the US had 8.5 times higher intention to use non-prescription antibiotics (95% CI [2.56–28.02], *p* < 0.001). Individuals with a higher preference to socialize in Spanish (lower Language Use subscale scores) expressed a higher intention to use non-prescribed antibiotics from any source (overall) in the future compared to those that had a lower preference to socialize in Spanish (higher Language Use subscale scores) (OR 0.61 95% CI [0.39–0.96], *p* = 0.031). Across all sources, prior non-prescription antibiotic use was a very strong predictor of the intention to use non-prescription antibiotics, with patients who reported previous non-prescription antibiotic use having over 12.3 times more intention to use non-prescription antibiotics in the future than patients who did not report prior non-prescription antibiotic use (95% CI [6.6–23.1], *p* < 0.001) (Table 8).

## 4. Discussion

This study investigated the effects of acculturation and subjective norms on Hispanic individuals’ intentions to use antibiotics without a prescription from the following sources: (1) markets in the United States (illegal), (2) other countries, (3) leftovers from previous prescriptions, and (4) friends/relatives. Our results underscore the alarmingly high proportion of Hispanic patients that have reported non-prescription antibiotic use in the past (47%) or intended to use them in the future (54%). We found that lower acculturation (i.e., language use and ethnic and social relations) and subjective norms favoring non-prescription antibiotic use were associated with higher patient intentions to use non-prescription antibiotics in the future. Simultaneously, healthcare system obstacles (i.e., high doctor visit costs and language barriers at the clinics) were associated with higher intended non-prescription antibiotic use.

Individuals of Hispanic heritage are one of the fastest-growing and largest foreign-born ethnic groups and are estimated to represent 25% of the entire US population by 2050 [22]. Across all sources, Hispanic patients born outside the US had nearly 8.5 times more intention to use non-prescribed antibiotics in our study. Similarly, studies in the US, Australia, and the United Kingdom have shown that immigrants may continue to practice self-medication behaviors that were common in their home countries, including using antibiotics without a prescription, for familiarity, convenience, sociocultural, and financial reasons [23,24,25]. Thus, it is imperative to understand the sociocultural factors that contribute to non-prescription antibiotic use to prevent this potentially unsafe practice. In addition, our results showed that patients’ prior non-prescription antibiotic use in the past year was a strong predictor of the intention to use non-prescription antibiotics in the future across all sources (OR 6.26 to 12.32, *p* < 0.001). These collective results pose an opportunity to develop antibiotic stewardship messaging based on the emerging recognition of the role of acculturation and social norms on non-prescription antibiotic use [26]. Healthcare professionals and health educators can promote safe antibiotic use as a social norm during patient–clinician counseling while also providing information on the individual-level harms and risks associated with antibiotic use, including *Clostridium difficile* infection, adverse effects, or drug interactions [26].

This research complements a growing body of literature on the association(s) between acculturation, subjective norms, and health behavior in Hispanic populations. Most prior research on these associations has focused on other health outcomes, including postpartum depression, nutrition, exercise, obesity, and cardiovascular disease (CVD), rather than inappropriate antibiotic use [11,13,27,28]. In this study, acculturation and subjective norms played an important role in Hispanic patients’ decisions to use non-prescription antibiotics. Specifically, we found correlations between lower acculturation (language use and ethnic and social relations subscale scores) and higher patient intentions to use non-prescribed antibiotics in the future, which is a novel finding. Similar to our results, a study in Texas found that less acculturated (by generational scores) Hispanic individuals were more likely to purchase antibiotics across the US–Mexico border, presumably without a prescription [11]. The lack of studies exploring the effects of acculturation on antibiotic use warrants further investigation into other ethnic groups across the US and other countries.

Our findings also highlight some specific healthcare barriers, including the high costs of doctor visits, long clinic waits (e.g., to schedule appointments or during doctor visits), and a lack of health insurance or health coverage, which impact Hispanic patients’ decisions to use non-prescription antibiotics [3,12]. Patients who experience the burden of high costs during a doctor’s visit also had three times higher intended non-prescription antibiotic use from the US and friends and relatives in comparison to patients that did not report high costs during a doctor’s visit as a problem. Another study also found that individuals without health insurance were over three times more likely to purchase antibiotics outside the US, presumably without a prescription [11]. In our previous qualitative study, high copayments (for a doctor’s visit and subsequent prescription medications), regardless of patients having healthcare coverage, drove some patients to seek informal medical advice and source non-prescription medications using their social networks [12]. Future research should leverage and promote appropriate antibiotic use as a social norm for Hispanic patients with and without healthcare coverage [13,29]. Specifically, engaging Hispanic communities with individuals whom they trust, such as community pharmacists and community healthcare workers (i.e., “promotoras”), in community stewardship interventions can help patients navigate the complex healthcare system [12,29]. A comprehensive approach that improves access to primary care may reduce non-prescription antibiotic use [12,29]. Moreover, antimicrobial stewardship programs administered by multidisciplinary teams [30] in hospital settings have led to beneficial clinical and economic impacts [31,32]. Therefore, implementing stewardship programs in outpatient settings could lead to similar outcomes, such as reducing inappropriate antibiotic use and limiting antimicrobial resistance.

Our study also identified the language barriers that Hispanic individuals may face during a doctor’s visit. Patients reporting language barriers as a problem during their healthcare visit reported three times more intended non-prescription antibiotic use from leftover sources. Hispanic patients’ negative healthcare experiences can have detrimental consequences. For instance, a recent Pew research study showed that approximately 50% of Hispanic Americans had negative healthcare experiences and difficulties getting needed healthcare, and 30% of Hispanic adults reported having to “speak up” (voice their concerns) to their doctors to get appropriate care [33]. For patients experiencing language barriers or limited English proficiency, this could be particularly discouraging, promoting alternative medical-seeking behaviors [12]. Similarly, compared to bilingual or English-only speaking Hispanic individuals surveyed, about 81% of the Spanish-speaking adults preferred seeing Spanish-speaking healthcare providers [33]. Addressing language barriers with language-concordant healthcare initiatives is important to mitigate communication pitfalls in medical care and has been shown to improve health outcomes [34].

Our study has certain limitations. First, our study does not compare Hispanic ethnic subgroups, and these communities encompass diverse cultures, backgrounds, and experiences. However, according to the US census and the Texas Demographics Center, Mexicans are the largest ethnic subgroup, representing 62% of all Hispanic people living in the US and 83% of all Hispanic people in Texas; thus, the Hispanic patients in this survey may represent the largest US Hispanic demographic subgroup [35,36]. Second, we did not account for immigration history or generational status (e.g., we did not ask patients about their parents’ or grandparents’ ancestry or when people first came to the US). To adjust for this, we calculated the median years lived in the US, but this factor was not a significant predictor of patient intentions to use non-prescription antibiotics. Third, the SASH scale may not account for all aspects of acculturation. Additionally, the SASH scale does not have any measures regarding the cultural context surrounding where study participants received their care (e.g., clinics and pharmacies). Nevertheless, the SASH scale has been found to have both high internal consistency and validity in measuring the language, media, and ethnic and social relations aspects of acculturation in many studies across a wide array of Hispanic subgroups [13,20,28]. Lastly, a social desirability response bias may have occurred despite our best efforts to phrase questions neutrally. Thus, the true prevalence rate of non-prescription antibiotic use may be underestimated, because patients may have had concerns about the legality or otherwise disclosing these behaviors or participating in the survey.

In summary, our results indicate that lower acculturation and subjective norms favoring non-prescription antibiotic use were associated with higher Hispanic patient intentions to use non-prescription antibiotics in the future. In addition, healthcare system obstacles, such as the high costs of doctor visits and language barriers, were associated with a higher intended non-prescription antibiotic use among Hispanic patients.

In conclusion, this study adds value to the scientific literature on the association(s) between acculturation, subjective norms, and health behavior in Hispanic populations. Reducing non-prescription antibiotic use in Hispanic communities in the US will require a multifaceted approach considering the sociocultural and healthcare barriers that influence non-prescription antibiotic use. Future stewardship interventions can leverage social and cultural factors to promote appropriate antibiotic use normative behaviors in Hispanic communities to reduce adverse health effects and antimicrobial resistance.

## Figures and Tables

**Figure 1 antibiotics-12-01419-f001:**
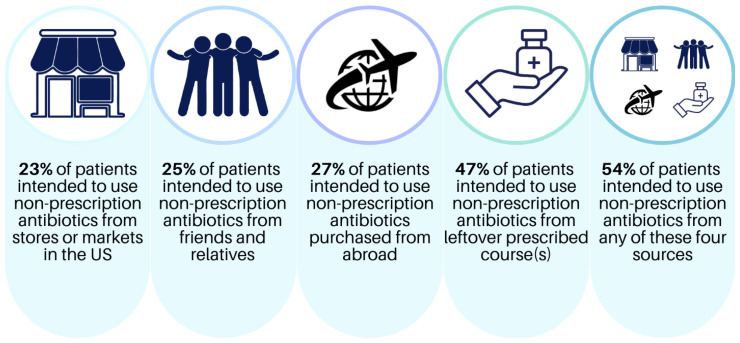
Prevalence of the intention to use non-prescription antibiotics in Hispanic patients surveyed (N = 263).

**Table 1 antibiotics-12-01419-t001:** Patient sociodemographic characteristics.

Characteristic	Total (N = 263)
Median age (y) (range)	51 (20–80)
No. (%) of female respondents	194 (73.8)
No. (%) of respondents with education level	
Less than high school	84 (31.9)
High school or GED	104 (39.5)
Some college and above	75 (28.5)
No. (%) of respondents with insurance status	
Private or Medicare	74 (28.1)
Medicaid or county financial assistance program *	173 (60.8)
Self-pay	16 (6.1)
No. (%) of patients attending Healthcare system	
Private	41 (15.6)
Public	222 (84.4)
No. (%) of patients attending clinic type	
Continuity clinic	102 (38.8)
Emergency Department	41 (15.6)
Walk in Clinic	120 (45.6)
No. of respondents with income/total no. of respondents (%)	
<$20,000	127 (48.3)
≥$20,000 but <$40,000	58 (22.1)
≥$40,000 but <$60,000	11 (4.2)
≥$60,000 but <$100,000	8 (3.0)
≥$100,000	6 (2.3)
Don’t know/prefer not to say	53 (20.2)
No. (%) of questionnaires completed in Spanish	155 (58.9)
No. (%) of respondents born in the United States/Other	
United States	81 (30.8)
Other ^†^	182 (69.2)
Median years lived in the United States for the respondents born in other countries (y) (range) (n = 182)	23 (0–58)
No. (%) of respondents reporting non-prescription antibiotic use	
Reported prior non-prescription use	123 (46.8)
No. (%) Health Literacy ^§^	
Adequate Health Literacy	198 (75.3)
Inadequate Health Literacy	65 (24.7)

* County financial assistance program includes those who have benefits from the county allowing access to public clinic providers at either very low cost or no cost. ^†^ Includes 1 Columbia, 1 Costa Rica, 6 Cuba, 1 Dominican Republic, 14 El Salvador, 6 Guatemala, 15 Honduras, 131 Mexico, 2 Nicaragua, 1 Panama, 1 Peru, and 3 Venezuela (countries are listed in alphabetical order). ^§^ Calculated using the three questions from the Brief Health literacy Screen measure [18,19].

**Table 2 antibiotics-12-01419-t002:** Acculturation by subscale means, interquartile range, and internal consistency ^†^.

Acculturation Subscales	Intended Use from Any Source		Cronbach’s Alpha
	Yes (n = 95) Median (IQR *)	No (n = 167) Median (IQR *)	
Language Use Subscale Score	2.0 (1.2–3.4)	2.0 (1.4–3.2)	0.939
Media Subscale Score	3.0 (1.7–4.0)	3.0 (1.0–4.0)	0.969
Ethnic Social Relations Subscale Score	2.3 (2.0–2.7)	2.3 (2.0–3.0)	0.817
Total Acculturation (Overall, Aggregate Score)	2.5 (1.6–3.4)	2.4 (1.7–3.3)	0.939

* Interquartile Range. ^†^ The Short Acculturation Scale for Hispanics is comprised of three subscales (Language Use, Media, and Ethnic Social Relations). Scores range from 1 (all Latinos/Hispanics) to 5 (all non-Latinos/Hispanics). Higher score indicates higher levels of acculturation [20].

**Table 3 antibiotics-12-01419-t003:** Subjective norms (N = 263).

	Total No. (%)
How many of your friends or relatives use antibiotics without contacting a doctor?	
None/Don’t Remember/Don’t Know	105 (39.9)
Some	98 (37.3)
About Half	15 (5.7)
Most	33 (12.5)
All	12 (4.6)
How many of your friends or relatives get antibiotics from another country?	
None/Don’t Remember/Don’t Know	123 (46.8)
Some	97 (36.9)
About Half	15 (5.7)
Most	28 (10.6)

## Data Availability

The data presented in this study are available on request from the corresponding author. The data are not publicly available due to privacy and ethical restrictions.

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
