# Peer review of "Acculturation and Subjective Norms Impact Non-Prescription Antibiotic Use among Hispanic Patients in the United States"

_antibiotics, 2023, doi:10.3390/antibiotics12091419_

Round 1

Reviewer 1 Report

The author investigates 'acculturation and subjective norms can impact health behaviors'. The author is supposed to associate the study with behavioral change theory. The behavioral change theory could explain the phenomenon and provide insight for designing (future stewardship) interventions for improving behavior. This aspect will improve the introduction, discussion, and conclusion sections.

Methods:

There is inconsistency: a validated acculturation scale in the abstract and survey instrument in the methods (p11, line 352). The author should add the reference for a validated acculturation scale (p1, line 24) or the validation process of the survey instrument.

The variety of respondents shows representativeness. How did the author choose the respondent? How does the author distribute and collect the data? How about the response rate?

Results:

Is Figure 1 - a part of Table 1?

What '(1 reference)' means in the Table 5-9 and Appendix C?

Only 6% of the respondent self-pay, but the high costs of doctor visits is a determinant of using antibiotic without a prescription. The author should discuss this finding more specifically, such as from the point of view of Hispanic socio-culture or any behavior change theory.

Discussion

What is the limitation of the study?

Author Response

Please, see the attachment for our detailed responses to Reviewer 1's suggestions. Thank you.

Reviewer 2 Report

Overall, the manuscript is well-written and provides valuable insights into the sociodemographic factors involved in the misuse of antibiotics among the Hispanic population. However, I do have some concerns that should be addressed to enhance the manuscript's quality:

1.     In the introduction section, it is crucial to include data about the correlation between Hispanic populations and other ethnic groups in terms of antibiotic misuse. Providing a comparative context will strengthen the manuscript's relevance and help readers better understand the specific challenges faced by the Hispanic population.

2.     I strongly suggest moving the materials and methods section up in the manuscript. By doing so, the flow of the paper will be improved, and readers will have a clear understanding of the study design and methodology right from the beginning.

3.     The patient recruitment process needs to be further elaborated. Specifically, more information should be provided regarding the type of outpatient clinic waiting room (e.g., Family Physician or Specialist setting), and whether the participants were volunteers or selected through a specific sampling method. Additionally, it is essential to clarify whether the survey instrument used in the study was validated and, if so, describe the validation process.

4.     In the discussion section, it is essential to address the correlation between antibiotic misuse and its impact on health and infections within the Hispanic population. By discussing the potential consequences of inappropriate antibiotic use, the manuscript will underscore the importance of addressing this issue for public health and patient outcomes.

On a positive note, I commend the authors for ensuring that the discussion and conclusion are in line with the study's results, providing a coherent and insightful interpretation of the findings. The references used in the manuscript appear relevant to the aim and content of the study, contributing to the overall strength of the paper.

In conclusion, addressing the mentioned concerns will significantly enhance the manuscript and strengthen its contribution to the field of antimicrobial stewardship among the Hispanic population.

Author Response

Please, see the attachment for our detailed responses to Reviewer 2's suggestions. Thank you.

Reviewer 3 Report

Overall the paper quality is good. In abstract, i suggest to use statistics for highlighting the significance result. Introduction is fine. Study limiting is missing so i suggest to add the limitation. 

Is there any criteria to include the sample?

How the questions are validated? its not clear in the paper?

In usa, where without pharmacist or medical order, antiobiotics are not allow to prescribe than how these patients taken from market? highligh the sources of market?

Conclusion is adequate.

Author Response

Please, see the attachment for our detailed responses to Reviewer 3's suggestions. Thank you.

Reviewer 4 Report

Very interesting topic for potential readers of the Journal. However, some comments are made in favor of improving the current version of the manuscript.

.- Title. Maybe very long. Evaluate redoing looking for it to be more attractive, e.g. “Non-prescription antibiotic use among Hispanic population of U.S, causes and consequences”

.- Abstract, ok.

.- Keywords. 8 words, perhaps it is excessive. Consider eliminating terms such as “Subjective Norms”, “Socio-cultural Factors” or “Social Norms”.

 .- Introduction. The last paragraph should be included in the Material and Methods section.

.- Material and Methods. It is usually included before the results section. Although it is mentioned, it is necessary to know the estimate of the sample size.

.- Results. It would be interesting to briefly explain what “Medicaid” is.

.- Discussion. Perhaps starting this section by directly mentioning “Our results…” is a bit abrupt. It is usual in this section, one or two introductory sentences on the subject of the manuscript.

Second paragraph refers to the Hispanic population in the U.S. It is missing previously mentioning "non-prescription antibiotic use".

Page 4. The expressions “our findings” and “we found” are repeated twice. Evaluate using synonymous expressions.

When it says “We found that patients who experience the burden of high costs during the doctor’s visit also had three times higher…”. To explain better.

Page 5. Twice the paragraph begins with “Our study...”. "Healthcare experiences" is repeated seven times. Review.

.- Conclusions. They are very short. A summary of the main conclusions obtained in this work is missing. Review.

.- References. Of the 34 citations provided, 19 (54.2%) are recent (that is, five years or less from its publication). Evaluate if it were possible to include any additional recent citations.

It would be a good idea to check all the references. There are several ones in which only "one author et al" appears, eg citations 3, 11 and 12. Review the Journal's regulations.

 .- Figures and Tables.

 Tables. Nine tables are presented. Maybe they are excessive. If they are explained in the text, perhaps some could be removed and/or included as “Supplementary Material”.

Figure 1. It’s very interesting. It is very visual, congratulations! Evaluate delete the absolute values ​​(60/263), leaving only the percentages. Consider arranging them in increasing order and perhaps making the circle larger in relation to the percentage.

A good english literary style is appreciated. Congratulations!

Author Response

Please, see the attachment for our detailed responses to Reviewer 4's suggestions. Thank you.

Round 2

Reviewer 2 Report

The quality of the manuscript significantly improved after the author's revision. 

Lastly, I suggest implementing the discussion by highlighting the importance of antimicrobial stewardship and adequate antibiotic prescription in healthcare facilities by providing recent literature evidence data.

I would suggest some citations:

https://doi.org/10.3390/ijerph20020996

https://doi.org/10.1128/cmr.18.4.638-656.2005

https://doi.org/10.1186/s13756-019-0471-0 

A minor English language and style check is required.

After these minor revisions, the manuscript could be accepted for publication.

A minor English language and style check is required.

Author Response

Thank you for the compliment. We agree with the reviewer’s suggestion to implement our discussion section by highlighting the importance of antimicrobial stewardship in diverse community/outpatient settings (i.e., the population we studied). To address the reviewer’s suggestion, we have added a section with recent citations to our discussion (lines 404-408) as follows,

“Moreover, antimicrobial stewardship programs administered by multidisciplinary teams [1] in hospital settings have led to beneficial clinical and economic impacts [2, 3]. Therefore, implementing stewardship programs in the outpatient settings could lead to similar outcomes, such as reducing inappropriate antibiotic use and limiting antimicrobial resistance.”

Lastly, we have checked the manuscript for grammar and style and find it aligned with MDPI’s and Antibiotics’ guidelines for authors.

Citations:

  1. MacDougall, C.; Polk, R.E. Antimicrobial stewardship programs in health care systems. Clin Microbiol Rev 2005, 18, 638-656, doi:10.1128/CMR.18.4.638-656.2005.
  2. Nathwani, D.; Varghese, D.; Stephens, J.; Ansari, W.; Martin, S.; Charbonneau, C. Value of hospital antimicrobial stewardship programs [ASPs]: a systematic review. Antimicrobial Resistance & Infection Control 2019, 8, 35, doi:10.1186/s13756-019-0471-0.
  3. Albano, G.D.; Midiri, M.; Zerbo, S.; Matteini, E.; Passavanti, G.; Curcio, R.; Curreri, L.; Albano, S.; Argo, A.; Cadelo, M. Implementation of A Year-Long Antimicrobial Stewardship Program in A 227-Bed Community Hospital in Southern Italy. International Journal of Environmental Research and Public Health 2023, 20, 996.